# Mapping multimorbidity in individuals with schizophrenia and bipolar disorders: evidence from the South London and Maudsley NHS Foundation Trust Biomedical Research Centre (SLAM BRC) case register

Rebecca Bendayan [1,2] Zeljko Kraljevic,[1] Shaweena Shaari,[2] Jayati Das-Munshi,[3] Leona Leipold [2] Jaya Chaturvedi,[1] Luwaiza Mirza,[2] Sarah Aldelemi,[2] Thomas Searle,[1] Natalia Chance,[2] Aurelie Mascio,[1] Naoko Skiada,[1] Tao Wang,[1] Angus Roberts,[1,2] Robert Stewart,[2,3] Daniel Bean,[1,4] Richard Dobson[1,2,5]

For numbered affiliations see end of article.

**Correspondence to**
Dr Rebecca Bendayan;
rebecca.bendayan@kcl.ac.uk

## ABSTRACT

**Objectives** The first aim of this study was to design and develop a valid and replicable strategy to extract physical health conditions from clinical notes which are common in mental health services. Then, we examined the prevalence of these conditions in individuals with severe mental illness (SMI) and compared their individual and combined prevalence in individuals with bipolar (BD) and schizophrenia spectrum disorders (SSD).

**Design** Observational study.

**Setting** Secondary mental healthcare services from South London

**Participants** Our maximal sample comprised 17 500 individuals aged 15 years or older who had received a primary or secondary SMI diagnosis (International Classification of Diseases, 10th edition, F20-31) between 2007 and 2018.

**Measures** We designed and implemented a data extraction strategy for 21 common physical comorbidities using a natural language processing pipeline, MedCAT. Associations were investigated with sex, age at SMI diagnosis, ethnicity and social deprivation for the whole cohort and the BD and SSD subgroups. Linear regression models were used to examine associations with disability measured by the Health of Nations Outcome Scale.

**Results** Physical health data were extracted, achieving precision rates (F1) above 0.90 for all conditions. The 10 most prevalent conditions were diabetes, hypertension, asthma, arthritis, epilepsy, cerebrovascular accident, eczema, migraine, ischaemic heart disease and chronic obstructive pulmonary disease. The most prevalent combination in this population included diabetes, hypertension and asthma, regardless of their SMI diagnoses.

**Conclusions** Our data extraction strategy was found to be adequate to extract physical health data from clinical notes, which is essential for future multimorbidity

## Strengths and limitations of this study

► We designed and implemented a data extraction strategy which showed high performance rates and allowed us to unlock data from 21 physical health conditions from around 15 m clinical documents with free text.

► We mapped how these health conditions are distributed across sex, age, ethnicity, social disadvantage and severe mental illness (SMI) diagnoses in a sample of 17 500 patients from one of Europe's largest providers of secondary mental healthcare, serving a geographical catchment of approximately 1.32 million residents in London.

► We investigated the association between multimorbidity (two or more conditions) and disability using the Health of the Nation Outcome Scale which is commonly collected in secondary mental health services in the UK.

► This study focuses on a cohort of individuals with SMI which limits direct comparisons with other mental health conditions and/or general population.

► Although some of the most frequent physical comorbidities were extracted, some specific health conditions required to compute standard comorbidity scores (eg, Charlson and Exlihauser comorbidity indexes) were not included in this study.

research using text records. We found that around 40% of our cohort had multimorbidity from which 20% had complex multimorbidity (two or more physical conditions besides SMI). Sex, age, ethnicity and social deprivation were found to be key to understand their heterogeneity and their differential contribution to disability levels in this population. These outputs have direct implications for researchers and clinicians.

## INTRODUCTION

Two-thirds of the deaths in individuals with severe mental illness (SMI) are potentially explained by the increased risk of multimorbidity in this population.[1–4] However, multimorbidity research in this population is still scarce[5] compounded by the limited availability of physical health data in SMI samples, increased non-response rates in surveys,[6] and physical health information in secondary mental healthcare data primarily hidden in free text fields.

Most research to date on physical health in SMI populations has focused on cardiometabolic risk factors, which are considered leading contributors to cardiovascular diseases in individuals with SMI,[7–10] or specific conditions such as immune-mediated inflammatory diseases (eg, inflammatory bowel diseases, psoriasis),[11–13] multiple sclerosis, epilepsy and migraine.[14 15] This condition-specific vision limits our understanding of multimorbidity in SMI and studies that consider a larger number of conditions are needed. However, there are only a few studies which have considered multiple health conditions.[4 16 17] Woodhead et al[4] showed an increased risk in multimorbidity in SMI patients, but only found epilepsy to be more prevalent as an individual condition. Kugathasan et al[17] investigated combinations of diseases in schizophrenia at organ system level and found that 31% had complex multimorbidity with the most prevalent pairs including neurologic-endocrine, neurologic-respiratory and neurological viral. Similarly, epilepsy and arthritis was one of the most prevalent combinations in Dorrington et al.[16] Although these studies included multiple health conditions and confirmed the need of investigating further multimorbidity in SMI, they are still not comparable to multimorbidity studies in general populations[18–20] and they did not investigate potential differences between individuals with schizophrenia spectrum disorders (SSD) and bipolar disorders (BD). Understanding different multimorbidity combinations between those groups could contribute to the ongoing debate around potential underlying biological mechanisms.[21–23] Ultimately, SSD and BD have been established as significant drivers of disability[24] and deficits in physical health have been implicated in the perpetuation of impairments in functional capacity and performance.[25 26] However, research into the relationship between multimorbidity and disability in SMI is limited.

Within this context, our first aim was to design and develop a suitable strategy to extract information on physical health conditions from free text mental health records data which could be easily replicated in future multimorbidity research using similar resources. Our second objective was to examine the prevalence of these conditions and their most prevalent combinations in SMI and any differences across relevant sociodemographic factors and across SMI diagnoses (SSD vs BD). Our third objective was to investigate the association of overall multimorbidity and specific physical health conditions with levels of disability measured using the Health of the Nation Outcome Scales (HoNOS).

## METHODS

### Setting and sample

Patient data were extracted via the Clinical Record Interactive Search (CRIS), a case register platform that contains de-identified mental healthcare electronic health record (EHR) data from the South London and Maudsley Trust National Health Service Foundation Trust (SLaM). SLaM is one of Europe's largest providers of secondary mental healthcare, serving a geographical catchment of approximately 1.32 million residents, and providing almost complete coverage of secondary mental healthcare provision to all age groups. Since 2007, fully electronic clinical records have been deployed in SLaM, and data from these are accessible via CRIS system which allows searching and retrieval of anonymised full records for over 500 000 cases currently represented in the system.[27]

Our sample (N=17 500) consisted of all individuals aged 15 years or older who had received a primary or secondary SMI diagnosis between 2007 and 2018 (International Classification of Mental and Behavioural Disorders 10th edition[28] codes F20-31). Given that one of our objectives was to compare SSD (F20-29) and BD (F30-31), individuals who over those 10 years of follow-up had diagnoses within both categories were excluded (n=804). Excluded individuals were more likely to be female, under the age of 35 at first SMI diagnosis recorded, Black ethnicity and have higher levels of social deprivation.

### Physical health conditions

#### Definitions and Information extraction

To maximise comparability, we sought to extract the following 21 physical health conditions representing chronic conditions commonly collected in multimorbidity studies using primary care data[18–20]: diabetes mellitus, heart failure, ischaemic heart diseases, hypertension, coronary arteriosclerosis, chronic obstructive pulmonary disease (COPD), asthma, chronic kidney disease (CKD), cerebrovascular accident, transient ischaemic attack (TIA), Parkinson's disease, multiple sclerosis, epilepsy, migraine, atrial fibrillation, chronic sinusitis, inflammatory bowel disease, chronic liver diseases, psoriasis, eczema and arthritis. These were mapped to SNOMED codes where the top concept was the group identifier and then all direct children of that concept were examined and individually reviewed by two clinicians (online supplemental appendix 1). Physical health conditions were ascertained from data reported in text records from CRIS since 2007 until 1 August 2019 for each individual resulting in around 15 m documents.

To extract SNOMED concepts from free text we used MedCAT,[29] a medical concept annotation toolkit capable of named entity recognition linking (NER+L) with contextualisation. The base model used is described in Kraljevic et al,[29] and has shown very good performance (F1=0.90). In a first step, the base model was enriched with concept names from UMLS with the purpose of increasing recall and potentially catching all different name-forms for each concept. In a second step, MedCAT was trained in

**Table 1** MedCAT performance F1, precision and recall estimates for each physical health conditions

| Physical health condition | F1 | Precision | Recall |
|---|---|---|---|
| Diabetes mellitus | 0.98 | 0.99 | 0.98 |
| Heart failure | 0.97 | 0.97 | 0.96 |
| Ischaemic heart disease | 0.98 | 0.97 | 0.99 |
| Hypertensive disorder, systemic arterial | 0.97 | 0.97 | 0.96 |
| Chronic obstructive lung disease | 0.94 | 0.97 | 0.92 |
| Asthma | 1.00 | 1.00 | 1.00 |
| Chronic kidney disease | 1.00 | 1.00 | 0.99 |
| Cerebrovascular accident | 0.96 | 0.94 | 0.98 |
| Transient ischaemic attack | 0.91 | 0.82 | 1.00 |
| Parkinson's disease | 0.94 | 0.88 | 1.00 |
| Multiple sclerosis | 1.00 | 1.00 | 1.00 |
| Epilepsy | 0.93 | 1.00 | 0.85 |
| Migraine | 1.00 | 1.00 | 1.00 |
| Atrial fibrillation | 0.98 | 1.00 | 0.96 |
| Chronic sinusitis | 0.98 | 0.97 | 1.00 |
| Inflammatory bowel disease | 0.96 | 1.00 | 0.92 |
| Chronic liver disease | 1.00 | 1.00 | 1.00 |
| Psoriasis | 1.00 | 1.00 | 1.00 |
| Eczema | 0.94 | 1.00 | 0.88 |
| Arthritis | 1.00 | 1.00 | 1.00 |

an unsupervised fashion on all the available documents to increase precision. In a third step, all the free text was annotated for the chosen SNOMED concepts. For each condition, 300 documents were randomly extracted, which resulted in a total of 6300 annotated documents.

### Annotation of physical health conditions

To ensure consistent, high-quality gold standard and training data, we developed annotation guidelines based on series of iterative discussions including clinical and technical expertise. These guidelines, available on request, were piloted and refined in preliminary stages. A relevant instance was defined as a mention of a physical health condition experienced by the patient and not negated. Each MedCAT detection was first validated as either correct/wrong—meaning the portion of text that was detected by MedCAT was either a correct/wrong detection of the relevant concept. Correct detections were further annotated with contextual annotations (or meta-annotations) for 'Diagnosis' and 'Status'. Diagnosis was used to determine if the detected concept is a patient related diagnosis, and Status if the detected concept is affirmed. Eight annotators were trained for this task and given the same instructions. MedCAT-trainer[30] was used to facilitate manual annotations and each document was double annotated. Disagreements between annotators were further evaluated and resolved by a third annotator.

### Training and validation

Once the dataset was annotated it was split into a training and validation set. For NER+L, 70% of the dataset was used for training and 30% for validation. For meta-annotations, 80% was used for training and 20% for validation. Hyperparameter optimisation in both cases used a 10-fold cross validation on the training set.

### Sociodemographics

Extracted data included sex, age at SMI diagnosis (15–24, 25–34, 35–44, 45–54, 55–64, 65–74, 75+) and ethnicity (white British, Irish, black Caribbean (including mixed white and black Caribbean and any other black background), black African (including mixed white and black African), South Asian (Indian, Pakistani and Bangladeshi) and other). Index of Multiple Deprivation (IMD) was extracted as a measure of neighbourhood socioeconomic status at the level of the 2011 lower layer super output area (LSOA11; a standard postal unit with an average 1500 residents) corresponding to the individual's address at time of SMI diagnosis. Using the IMD, each LSOA11 is ranked from 1 (most deprived) to 32 844 (least deprived) based on seven Census-derived indicators, which was subsequently divided into quintiles.[31]

### Disability

Disability was measured using Health of the Nation Outcome Scales (HoNOS)[32] which is a clinician-rated tool developed to measure health and social functioning. It includes 12 subscales: agitated behaviour; non-accidental self-injury; problem drinking or drug taking; cognitive problems; physical illness problems; problems associated with hallucinations or delusions; problems associated with depression; other mental and behaviour problems; problems with relationships; problems with activities of daily living; problems with living conditions; and problems with occupation and activities.[32] Total adjusted HoNOS scores of individuals at the first SMI diagnosis recorded in CRIS, or closest to that time, were used in this study. Higher scores for HoNOS indicate higher levels of impairment in the individual's functioning.

### Statistical analyses

To explore the suitability of MedCAT for extracting these physical health conditions from this cohort (objective 1), inter-rater agreement estimates were computed and performance, precision and recall per condition were estimated.

To examine the prevalence of these conditions in SMI across relevant factors and compare the most prevalent multimorbidity combinations for individuals with BD and SSD (objective 2). Descriptive statistics were derived for all the variables. $\chi^2$ tests and Fisher's exact tests, with Bonferroni correction for multiple comparisons, were performed to explore associations and differences between BD and SSD.

To address our third objective (to investigate the association of multimorbidity and specific physical health with

**Table 2**  Sociodemographic characteristics and prevalence for physical health conditions for total cohort (N=17 500) and by SMI diagnosis

|  | Total | SSD | BD |
|---|---|---|---|
| N (%) | 17 500 | 13 019 (74.4) | 4481 (25.6) |
| Sex*** |  |  |  |
| Female | 8123 (46.4) | 5421 (41.6) | 2702 (60.3) |
| Male | 9374 (53.6) | 7596 (58.3) | 1778 (39.7) |
| Age at first SMI diagnosis*** |  |  |  |
| 15–34 | 7497 (42.8) | 5607 (43.1) | 1890 (42.2) |
| 35–44 | 3736 (21.3) | 2792 (21.4) | 944 (21.1) |
| 45–54 | 2783 (15.9) | 2057 (15.8) | 726 (16.2) |
| 55–64 | 1525 (8.7) | 1067 (8.2) | 458 (10.2) |
| 65+ | 1959 (11.2) | 1496 (11.5) | 463 (10.3) |
| Ethnicity*** |  |  |  |
| White British | 6243 (35.7) | 4008 (30.8) | 2235 (49.9) |
| Black Caribbean | 3182 (18.2) | 2799 (21.5) | 383 (8.5) |
| Black African | 2094 (12.0) | 1886 (14.5) | 208 (4.6) |
| South Asian | 549 (3.1) | 421 (3.2) | 128 (2.9) |
| Irish | 346 (2.0) | 240 (1.8) | 106 (2.4) |
| Other | 3846 (22.0) | 2822 (21.7) | 1024 (22.9) |
| Not stated | 1240 (7.1) | 843 (6.5) | 397 (8.9) |
| Index of Multiple Deprivation*** |  |  |  |
| 1 (less deprived) | 742 (4.2) | 412 (3.2) | 330 (7.4) |
| 2 | 1384 (7.9) | 887 (6.8) | 497 (11.1) |
| 3 | 3575 (20.4) | 2503 (19.2) | 1072 (23.9) |
| 4 | 7073 (40.4) | 5476 (42.1) | 1597 (35.6) |
| 5 (more deprived) | 4033 (23.0) | 3204 (24.6) | 829 (18.5) |
| Unknown | 693 (4.0) | 537 (4.1) | 156 (3.5) |
| No of conditions*** |  |  |  |
| No mentions | 10 468 (59.8) | 7540 (57.9) | 2928 (65.3) |
| One | 3733 (21.3) | 2888 (22.2) | 845 (18.9) |
| Two | 1795 (10.3) | 1429 (11.0) | 366 (8.2) |
| Three or more | 1504 (8.6) | 1162 (8.9) | 342 (7.6) |
| Physical conditions*** |  |  |  |
| Diabetes*** | 2686 (15.3) | 2208 (17.0) | 478 (10.7) |
| Hypertension*** | 2537 (14.5) | 2070 (15.9) | 467 (10.4) |
| Asthma | 1722 (9.8) | 1291 (9.9) | 431 (9.6) |
| Arthritis | 954 (5.5) | 702 (5.4) | 252 (5.6) |
| Epilepsy*** | 799 (4.6) | 652 (5.0) | 147 (3.3) |
| Cerebrovascular accident | 728 (4.2) | 573 (4.4) | 155 (3.5) |
| Eczema | 616 (3.5) | 479 (3.7) | 137 (3.1) |
| Migraine*** | 564 (3.2) | 372 (2.9) | 192 (4.3) |
| Ischaemic heart disease | 561 (3.2) | 435 (3.3) | 126 (2.8) |
| Chronic obstructive pulmonary disease | 476 (2.7) | 342 (2.6) | 134 (3.0) |
| Chronic kidney disease* | 279 (1.6) | 179 (1.4) | 100 (2.2) |
| Parkinson's disease | 266 (1.5) | 201 (1.5) | 65 (1.5) |
| Heart failure** | 222 (1.3) | 187 (1.4) | 35 (0.8) |
| Psoriasis | 179 (1.0) | 129 (1.0) | 50 (1.1) |
| Atrial fibrillation | 133 (0.8) | 100 (0.8) | 33 (0.7) |

Continued

**Table 2** Continued

| | Total | SSD | BD |
|---|---|---|---|
| Transient ischaemic attack | 130 (0.7) | 91 (0.7) | 39 (0.9) |
| Inflammatory bowel disease | 40 (0.2) | 25 (0.2) | 15 (0.3) |
| Multiple sclerosis | 32 (0.2) | 19 (0.1) | 13 (0.3) |
| Chronic liver disease | 22 (0.1) | 20 (0.2) | 2 (0.0) |
| Chronic sinusitis | 6 (0.0) | 6 (0.0) | 0 (0.0) |

*P<0.05, **p<0.01, ***p<0.001 for comparisons between BD and SSD groups
BD, bipolar disorder; SMI, severe mental illness; SSD, schizophrenia spectrum disorder.

levels of disability), we performed series of hierarchical linear regressions. Models were adjusted by age and sex (model 1), and then additionally adjusted by IMD (Model 2a) or SMI diagnosis (model 2b). All analyses were performed using R V.4.0.3 and RStudio V.1.3.1093.

### Patient and public involvement statement

When designing this project, the Data Linkage Service User and Carer Advisory Group was consulted and followed up presenting preliminary results. This is a well-established patient and public involvement group set up by the Biomedical Research Centre at SLaM.[33]

### RESULTS

### Interannotator agreement and model validation for data extraction

For each physical health condition, 300 documents were annotated to create a gold standard and training data specific to each condition. All 6300 instances across 21 health conditions were double annotated yielding an average interannotator agreement of 97% for NER+L, 82.70% for the meta-annotation Diagnosis and 78.08% for the meta-annotation Status. Precision, recall and F1 metrics of each modelled physical health condition are shown in table 1. Coronary arteriosclerosis was not extracted as the number of positive mentions was too small for training and validation. Overall meta-annotations performance results showed good performance for diagnosis and status (online supplemental table 1).

### Mapping of physical health conditions and comparison between SSD and BD

Our sample consisted of 17 500 individuals with SMI, of whom 74.4% were diagnosed with SSD and 25.6% with BD. A slight majority were male (53.6%), and most individuals had their first SMI diagnoses report under the age of 35 (42.8%). The white British group accounted for 35.7% of our sample, followed by black Caribbean (18.2%) and black African (12.0%). The South Asian and Irish groups were the smallest, with 3.1% and 2.0%, respectively (table 2). There were high levels of deprivation in the cohort, with over 60% falling into the lowest two national quintiles. Around 40% had at least one mention of a physical health condition and around 20% had two or more physical conditions. There were significant differences

between BD and SSD for most of the socio-demographic characteristics and number of physical health conditions (table 2 and figure 1). Individuals with SSD were more likely to be men, from ethnic minorities, living in more deprived neighbourhoods and had a higher number of physical health conditions recorded compared with those with BD.

The three most common physical health conditions recorded were diabetes, hypertension and asthma (15.3%, 14.5% and 9.8%, respectively), regardless of the specific SMI diagnoses (online supplemental figure 1) within SSD and BD. When we compared individuals with SSD and BD, we found that the top 10 most prevalent health conditions were similar between groups but diabetes (SSD 17% vs BD 10.7%), hypertension (SSD 15.9% vs BD 10.4%) and epilepsy (SSD 5.0% vs BD 3.3%) prevalence rates were slightly higher for individuals with

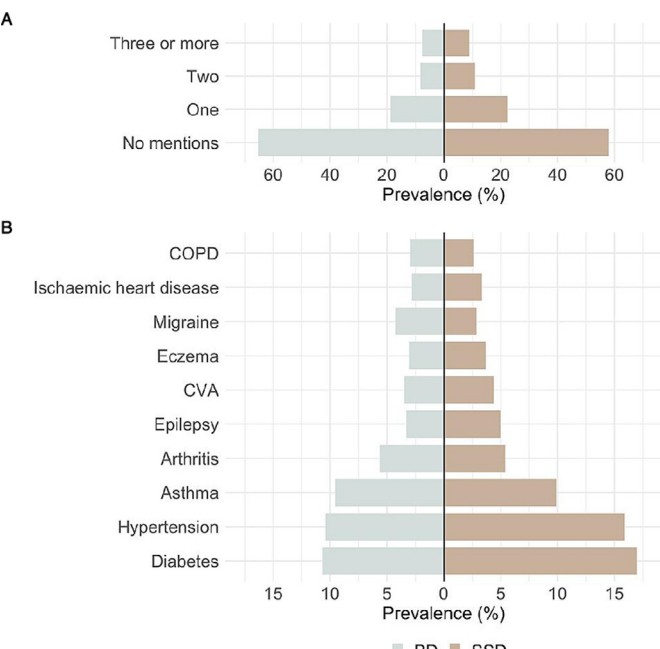

**Figure 1** Comparison of number of physical health comorbidities (A) and specific physical comorbidities (B) by SMI diagnosis. BD, bipolar disorder; COPD, chronic obstructive pulmonary disease; CVA, cerebrovascular accident; SMI, severe mental illness; SSD, schizophrenia spectrum disorder.

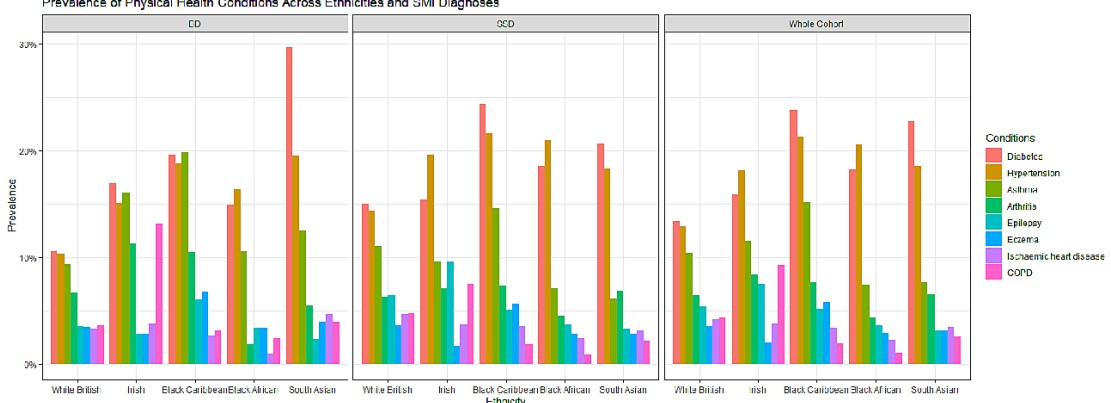

**Figure 2** Prevalence of the most prevalent physical health conditions across ethnicities within the SMI cohort and the SSD and BD subgroups. BD, bipolar disorder; COPD, chronic obstructive pulmonary disease; SMI, severe mental illness; SSD, schizophrenia spectrum disorder.

SSD while individuals with BD showed higher prevalence rates of migraine (BD 4.3% vs SSD 2.9%).

When we explored differences by sex in the whole cohort, we found that women were more likely to report hypertension, asthma, arthritis, eczema and migraine compared with men (online supplemental table 2). Within the individuals with SSD, women were found to be more likely to report higher rates of diabetes, CKD, heart failure and TIA. Within individuals with BD, sex differences were only found for asthma, arthritis, migraine and ischaemic heart disease. Women with BD were more likely to report asthma, arthritis and migraine while men with BD were more likely to report ischaemic heart disease.

With regard to differences across age groups, we found higher prevalence rates of diabetes, hypertension, arthritis, cerebrovascular accident, ischaemic heart disease, COPD, CKD, Parkinson's disease, heart failure, atrial fibrillation and TIA in individuals in older age ranges, while asthma and migraine were more prevalent in younger age ranges (online supplemental table 3), (online supplemental figure 2). We found similar results within individuals with SSD and BD (except for asthma and cerebrovascular accident in BD).

We found differences for ethnicity in individuals with diabetes, hypertension, asthma, arthritis, epilepsy, eczema, ischaemic heart disease, COPD and cerebrovascular accident (figure 2, online supplemental table 4). Individuals from black or South Asian minorities were more likely to show higher prevalence rates of diabetes and hypertension compared with those white British or Irish. Black Caribbean showed the highest prevalence rates of asthma or eczema among all other groups. Arthritis, COPD, epilepsy and IHD seem to be slightly more prevalent in White British or Irish, with epilepsy showing the highest prevalence rates among Irish. Similar trends were found within SSD and BD subgroups with diabetes rates higher in South Asians with BD (29.7%) compared with South Asians with SSD (20.7%); while diabetes rates in Black Caribbean with SSD (24.4%) were higher than in those with BD (19.6%). With regards to social deprivation, we found that individuals with diabetes, hypertension,

asthma, and COPD were more likely to be at higher levels of deprivation compared with those that did not have these specific conditions (online supplemental table 5).

## Multimorbidity combinations for the whole cohort and SSD and BD subgroups

Table 3 summarises the 10 most common physical comorbidities in patients with SMI, their prevalence, the mean number of comorbidities and the three most frequently associated comorbidities, for the total cohort and by SMI diagnosis. While there were no clear differences in the mean number of comorbidities by SMI diagnosis, the presence of one physical condition predisposed individuals to at least having one other condition. The mean number of comorbidities in the total cohort was 0.74 jumped to at least 2.20 in the presence of one of the 10 most common comorbidities. The three most commonly associated physical comorbidities remained relatively consistent by SMI diagnosis, with a few exceptions. The prevalence of associated comorbidities with epilepsy were lower in BD than in SSD; there is a fairly different comorbidity profile in migraine between SMI diagnoses; and, a lower rate of diabetes in individuals with comorbid BD and COPD (when compared with SSD). The most common combination of conditions included diabetes, hypertension and asthma, regardless of their SMI diagnoses. Most individuals with these combinations of conditions were also likely to have arthritis. Figures 3–5 show the most prevalent conditions for individuals with SMI and comorbid diabetes, hypertension and asthma, respectively.

## Association with disability

HoNOS descriptive statistics for the whole SMI cohort (Mean=10.40, SD=6.06) and for those with the 10 most common physical comorbidities are shown in table 4. Regression analyses showed that individuals with any of these conditions (except migraine) showed higher HoNOS scores, even after adjustments for age, sex, IMD or SMI diagnoses. We also examined whether simple and complex multimorbidity was associated with HoNOS total score and we found a strong positive association

**Table 3** Ten most prevalent conditions and associated comorbidities of the total cohort and by SMI diagnosis

| Condition | | Prevalence | Mean # of comorbidities | Three most frequent associated comorbidities | | |
|---|---|---|---|---|---|---|
| | Total | – | 0.74 | 1. Diabetes (15.3%) | 2. HTN (14.5%) | 3. Asthma (9.8%) |
| | SSD | – | 0.77 | 1. Diabetes (17.0%) | 2. HTN (15.9%) | 3. Asthma (9.9%) |
| | BD | – | 0.64 | 1. Diabetes (10.7%) | 2. HTN (10.4%) | 3. Asthma (9.6%) |
| Diabetes | Total | 15.3% | 2.35 | 1. HTN (42.1%) | 2. Asthma (16.9%) | 3. Arthritis (13.2%) |
| | SSD | 17.0% | 2.33 | 1. HTN (42.9%) | 2. Asthma (16.3%) | 3. Arthritis (12.7%) |
| | BD | 10.7% | 2.45 | 1. HTN (38.3%) | 2. Asthma (19.5%) | 3. Arthritis (15.5%) |
| Hypertension | Total | 14.5% | 2.50 | 1. Diabetes (44.5%) | 2. Asthma (16.3%) | 3. Arthritis (15.2%) |
| | SSD | 15.9% | 2.47 | 1. Diabetes (45.7%) | 2. Asthma (16.1%) | 3. Arthritis (14.7%) |
| | BD | 10.4% | 2.62 | 1. Diabetes (39.2%) | 2. Arthritis (17.3 %) | 3. Asthma (17.1%) |
| Asthma | Total | 9.8% | 2.27 | 1. Diabetes (26.3%) | 2. HTN (24.0%) | 3. Arthritis (11.9%) |
| | SSD | 9.9% | 2.27 | 1. Diabetes (27.9%) | 2. HTN (25.9%) | 3. Eczema (11.2%) |
| | BD | 9.6% | 2.29 | 1. Diabetes (21.6%) | 2. HTN (18.6%) | 3. Arthritis (15.1%) |
| Arthritis | Total | 5.5% | 2.78 | 1. HTN (40.5%) | 2. Diabetes (37.1%) | 3. Asthma (21.5%) |
| | SSD | 5.4% | 2.78 | 1. HTN (43.4%) | 2. Diabetes (39.9%) | 3. Asthma (19.9%) |
| | BD | 5.6% | 2.80 | 1. HTN (32.1%) | 2. Diabetes (29.4%) | 3. Asthma (25.8%) |
| Epilepsy | Total | 4.6% | 2.40 | 1.HTN (25.5%) | 2.Diabetes (24.8%) | 3.Asthma (21.4%) |
| | SSD | 5.0% | 2.42 | 1. HTN (27.3%) | 2. Diabetes (26.4%) | 3. Asthma (21.8%) |
| | BD | 3.3% | 2.31 | 1. Asthma (19.7%) | 2. Diabetes (17.7%) | 2. HTN (17.7%) |
| CVA | Total | 4.2% | 2.89 | 1. HTN (42.3%) | 2. Diabetes (38.6%) | 3. Asthma (15.4%) |
| | SSD | 4.4% | 2.83 | 1. HTN (42.8%) | 2. Diabetes (38.9%) | 3. Asthma (14.0%) |
| | BD | 3.5% | 3.09 | 1. HTN (40.6%) | 2. Diabetes (37.4%) | 3. Asthma (21.9) |
| Eczema | Total | 3.5% | 2.50 | 1. Asthma (32.5%) | 2. Diabetes (26.9%) | 3. HTN (23.1%) |
| | SSD | 3.7% | 2.45 | 1. Asthma (30.3%) | 2. Diabetes (28.4%) | 3. HTN (22.5%) |
| | BD | 3.1% | 2.64 | 1. Asthma (40.1%) | 2. HTN (24.8%) | 3. Diabetes (21.9%) |
| Migraine | Total | 3.2% | 2.20 | 1. Asthma (23.6%) | 2. Diabetes (20.0%) | 3. HTN (18.4%) |
| | SSD | 2.9% | 2.23 | 1. Diabetes (23.4%) | 2. Asthma (21.5%) | 3.HTN (20.2%) |
| | BD | 4.3% | 2.15 | 1. Asthma (27.6%) | 2. HTN (15.1%) | 3. Arthritis (14.1%) |
| Ischaemic heart disease | Total | 3.2% | 3.27 | 1. HTN (49.0%) | 2. Diabetes (43.3%) | 3. Asthma (20.3%) |
| | SSD | 3.3% | 3.28 | 1. HTN (51.0%) | 2. Diabetes (44.6%) | 3. Asthma (20.2%) |
| | BD | 2.8% | 3.24 | 1. HTN (42.1%) | 2. Diabetes (38.9%) | 3. Arthritis (22.2%) |
| COPD | Total | 2.7% | 3.22 | 1. HTN (39.7%) | 2. Diabetes (38.9%) | 3. Asthma (35.7%) |
| | SSD | 2.6% | 3.20 | 1. Diabetes (41.2%) | 2. HTN (39.2%) | 3. Asthma (35.7%) |
| | BD | 3.0% | 3.25 | 1. HTN (41.0%) | 2.Asthma (35.8%) | 3. Diabetes (32.8%) |

BD, Bipolar disorders; COPD, chronic obstructive pulmonary disease; CVA, cerebrovascular accident; HTN, hypertension; SMI, severe mental illness; SSD, schizophrenia Spectrum disorders.

minimally attenuated after adjustments. Similar sociodemographics and trends were found within BD and SSD groups (online supplemental tables 6–8). However, associations for diabetes, hypertension and ischaemic heart disease were fully attenuated after adjustments in the SSD group and associations for hypertension were also fully attenuated after adjustments in the BD group.

## DISCUSSION

The first objective of this study was to design and develop a suitable strategy to extract physical health conditions which could be easily replicated in future multimorbidity research using mental health EHRs. The natural language processing strategy using MedCAT provided very good performance estimates for all the conditions extracted, which supports its suitability to extract data on physical health conditions from mental health clinical notes. These findings are consistent with previous research which has used MedCAT to extract data from hospital settings.[34][35] This resource should help to facilitate and promote research on multimorbidity using mental health records, in general, and has the potential for direct replication in other mental health trusts which have already deployed CRIS platforms.

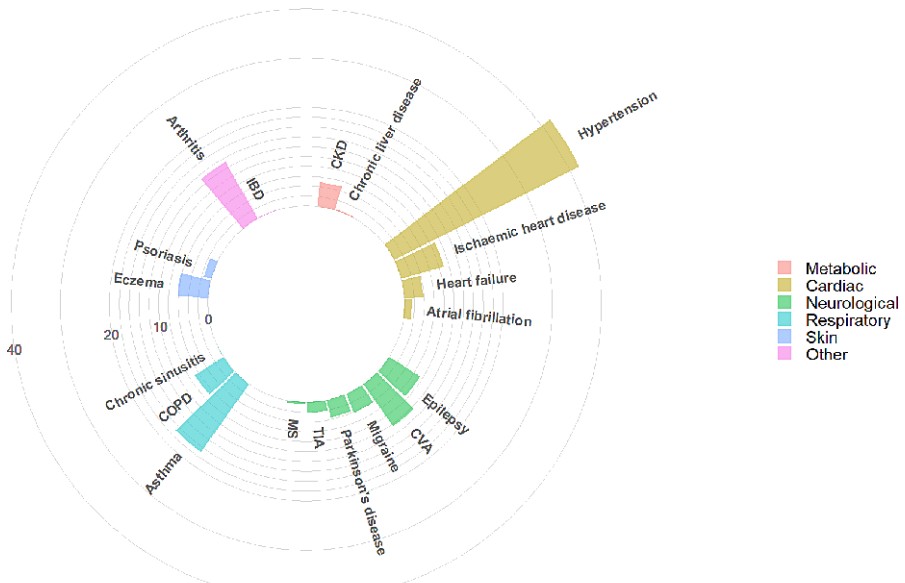

**Figure 3** Visualisation of most prevalent comorbidities in individuals with SMI and comorbid diabetes. CKD, chronic kidney disease; COPD, chronic obstructive pulmonary disease; CVA, cerebrovascular accident; IBD, inflammatory bowel disease; MS, multiple sclerosis; SMI, severe mental illness; TIA, transient ischaemic attack.

Our second objective was to examine the prevalence of these conditions in SMI individuals and compare the most prevalent multimorbidity combinations between individuals with BD and SSD. When we examined differences in socio-demographic variables by diagnosis, our findings were largely consistent with previous research,[3 36–39] although associations between ethnicity and BD are less established.[40] With regards to sociodemographic differences, we found that women with SSD were more likely

to have diabetes, CKD, heart failure and TIA compared with men with SSD; and women with BD were more likely to have asthma, arthritis and migraine compared with men with BD. Previous research in this population showed mixed results. Some studies found higher prevalence of hypertension in women with SSD[41] and diabetes in women in BD,[42] and others did not find relevant sex differences.[43] Our findings suggest that there could be an increased risk for diabetes and hypertension for females

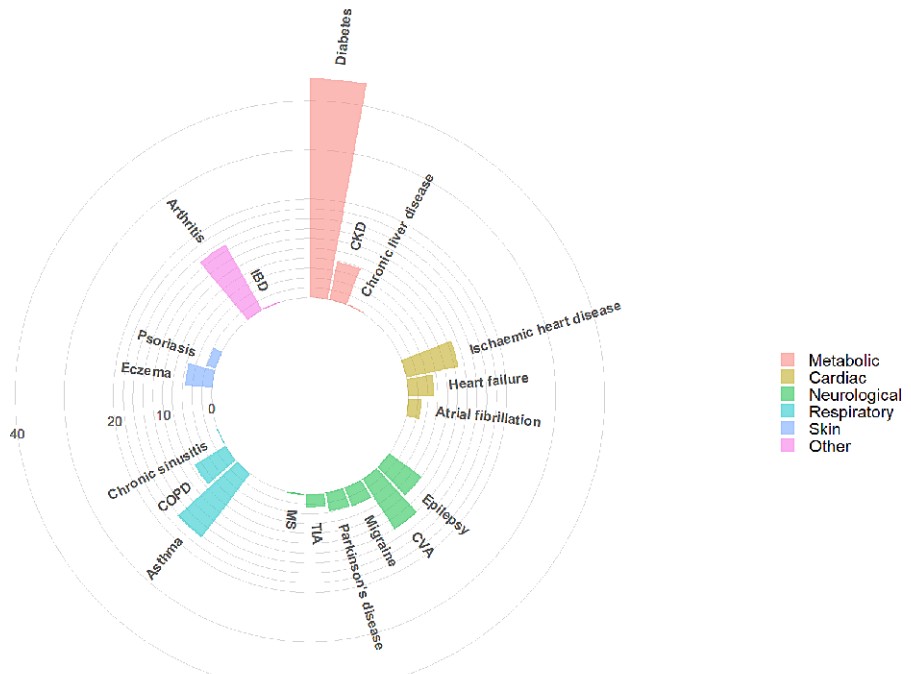

**Figure 4** Visualisation of most prevalent comorbidities in individuals with SMI and comorbid hypertension. CKD, chronic kidney disease; COPD, chronic obstructive pulmonary disease; CVA, cerebrovascular accident; IBD, inflammatory bowel disease; MS, multiple sclerosis; SMI, severe mental illness; TIA, transient ischaemic attack.

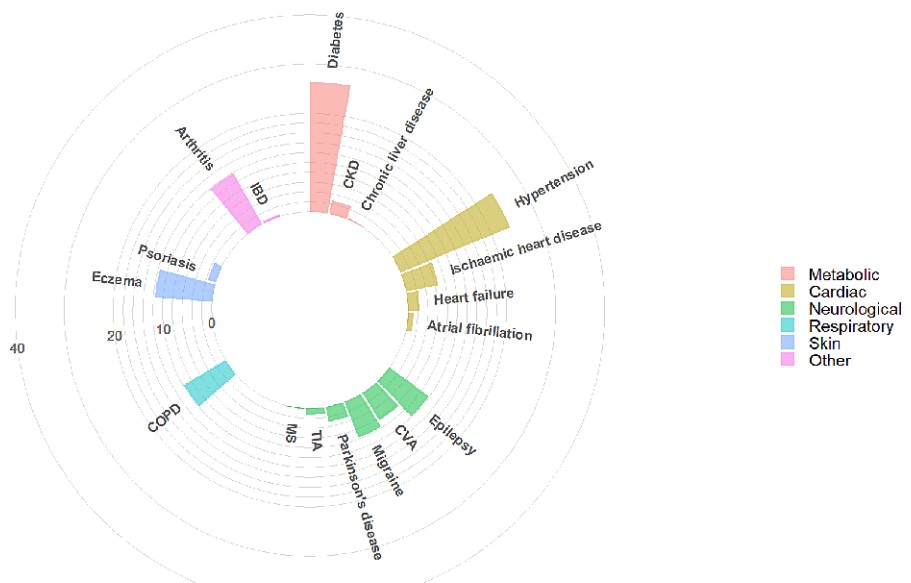

**Figure 5** Visualisation of most prevalent comorbidities in individuals with SMI and comorbid asthma. CKD, chronic kidney disease; COPD, chronic obstructive pulmonary disease; CVA, cerebrovascular accident; IBD, inflammatory bowel disease; MS, multiple sclerosis; SMI, severe mental illness; TIA, transient ischaemic attack.

with an SMI diagnosis, especially with SSD. Further research in this line is needed.

Ethnic differences were found for diabetes, hypertension, asthma, arthritis, epilepsy, eczema, ischaemic heart disease, COPD and cerebrovascular accident. Individuals from black or South Asian minorities were more likely to show higher prevalence rates of diabetes and hypertension compared with white British or Irish. Black Caribbean showed the higher prevalence rates of asthma or eczema among all other groups. Arthritis, COPD, epilepsy and IHD seem to be slightly more prevalent in white British or Irish, with epilepsy showing the highest prevalence rates among Irish. Similar trends were found within SSD and BD subgroups with diabetes rates higher in South Asians with BD compared with South Asians with SSD, while diabetes rates in black Caribbean with SSD were higher than in those with BD. These results largely mirror previous research in ethnicity.[44–50] When we examined social deprivation, individuals with diabetes, hypertension, asthma and COPD were more likely to report the highest levels of deprivation regardless of their SMI diagnoses. Similar to ethnicity, these results are also consistent with findings in the general population where higher levels of social deprivation are found in those with comorbid diabetes,[51 52] hypertension,[53] asthma[54] or COPD.[55]

Overall, in the whole SMI cohort, around 40% of the individuals had at least one mention of a physical health condition and close to 20% had two or more physical conditions, which could be labelled as complex multimorbidity. These findings provide evidence to support previous research suggestions about the increased probability of multimorbidity in this population.[3 17 56–59] Absolute numbers of physical health conditions were higher in patients with SSD than those with BD. Although direct

comparisons require caution, our findings partially contrast with previous reports of higher number of physical comorbidities in individuals with BD.[3 38 39 41] Overall, the top 10 most prevalent conditions in our SMI cohort were diabetes, hypertension, asthma, arthritis, epilepsy, cerebrovascular accident, eczema, migraine, ischaemic heart disease and COPD; and the most common combination of conditions included diabetes, hypertension and asthma, regardless of their SMI diagnoses. Moreover, those that had complex multimorbidity were also more likely to have cardiometabolic comorbidities such as diabetes and hypertension, which suggests that the cardiometabolic pathway might be one of the key explanatory mechanism underlying the association between physical multimorbidity and SMI.[60 61] Future research should explore further the potential independent contribution of this pathway when focusing on individuals with complex multimorbidity. Furthermore, arthritis was the most frequent subsequent comorbidity for those with diabetes, hypertension and/or asthma, however, for those with SSD and asthma, eczema slightly displaced arthritis in terms of prevalence. These findings might suggest that potential differences between SSD and BD phenotypes could be linked to underlying inflammatory pathways. Future research focusing on inflammatory biomarkers could be key to further our understanding of the potential differences between SSD and BD.

In addition, we examined the association between the top 10 most prevalent conditions and disability levels. We found that not only multimorbidity was clearly associated with higher levels of disability but having any of these specific conditions was associated with higher levels of disability, even after adjusting for age, sex, deprivation or SMI diagnoses. Similar results were found when we examined the associations between multimorbidity

**Table 4** Associations between specific comorbidities, multimorbidity and HoNOS scores.

| | HoNOS | Unadjusted | Model 1 | Model 2A | Model 2B |
|---|---|---|---|---|---|
| | Mean (SD) | B (95% CI) | B (95% CI) | B (95% CI) | B (95% CI) |
| Physical comorbidities | | | | | |
| Diabetes | 10.93 (6.08) | 0.652 (0.389 to 0.914)*** | 0.464 (0.198 to 0.730)*** | 0.485 (0.215 to 0.755)*** | 0.359 (0.094 to 0.625)** |
| Ref: No diabetes | 10.28 (6.05) | | | | |
| Hypertension | 10.99 (6.02) | 0.713 (0.446 to 0.980)*** | 0.401 (0.123 to 0.680)** | 0.368 (0.084 to 0.652)* | 0.297 (0.019 to 0.576)* |
| Ref: No hypertension | 10.27 (6.06) | | | | |
| Asthma | 11.05 (6.16) | 0.734 (0.417 to 1.051)*** | 0.806 (0.490 to 1.121)*** | 0.770 (0.450 to 1.091)*** | 0.804 (0.489 to 1.118)*** |
| Ref: No asthma | 10.31 (6.04) | | | | |
| Arthritis | 12.01 (6.19) | 1.728 (1.322 to 2.134)*** | 1.570 (1.156 to 1.984)*** | 1.558 (1.142 to 1.974)*** | 1.567 (1.155 to 1.980)*** |
| Ref: No arthritis | 10.28 (6.03) | | | | |
| Epilepsy | 11.40 (6.19) | 1.055 (0.596 to 1.514)*** | 1.066 (0.609 to 1.523)*** | 1.089 (0.628 to 1.549)*** | 0.982 (0.527 to 1.437)*** |
| Ref: No epilepsy | 10.35 (6.05) | | | | |
| CVA | 11.81 (6.22) | 1.487 (1.022 to 1.951)*** | 1.195 (0.728 to 1.662)*** | 1.180 (0.712 to 1.649)*** | 1.162 (0.697 to 1.627)*** |
| Ref: No CVA | 10.33 (6.04) | | | | |
| Eczema | 11.12 (6.19) | 0.753 (0.249 to 1.257)** | 0.838 (0.336 to 1.340)*** | 0.728 (0.220 to 1.237)** | 0.799 (0.299 to 1.299)** |
| Ref: No eczema | 10.37 (6.05) | | | | |
| Migraine | 10.33 (5.73) | −0.078 (−0.600 to 0.445) | 0.212 (−0.311 to 0.734) | 0.227 (−0.300 to 0.754) | 0.302 (−0.218 to 0.823) |
| Ref: No migraine | 10.40 (6.07) | | | | |
| Ischaemic heart disease | 11.64 (5.92) | 1.294 (0.766 to 1.822)*** | 0.864 (0.332 to 1.395)*** | 0.853 (0.318 to 1.387)** | 0.849 (0.320 to 1.379)** |
| Ref: No ischaemic heart disease | 10.35 (6.06) | | | | |
| COPD | 12.08 (5.78) | 1.733 (1.158 to 2.309)*** | 1.326 (0.745 to 1.908)*** | 1.336 (0.750 to 1.923)*** | 1.397 (0.818 to 1.977)*** |
| Ref: No COPD | 10.34 (6.06) | | | | |
| Multimorbidity | | | | | |
| No of comorbidities | | 0.487 (0.406 to 0.567)*** | 0.423 (0.339 to 0.507)*** | 0.424 (0.339 to 0.510)*** | 0.404 (0.320 to 0.488)*** |
| One or more comorbidities | 10.96 (6.08) | 1.053 (0.850 to 1.256)*** | 0.893 (0.685 to 1.101)*** | 0.895 (0.681 to 1.109)*** | 0.823 (0.615 to 1.031)*** |
| Ref: No comorbidities | 9.90 (5.99) | | | | |
| Two or more comorbidities | 11.34 (6.13) | 1.214 (0.973 to 1.455)*** | 1.021 (0.772 to 1.269)*** | 1.020 (0.768 to 1.272)*** | 0.968 (0.720 to 1.216)*** |
| Ref: Less than one comorbidities | 10.12 (6.01) | | | | |

Models were adjusted by age and sex (model 1), and then additionally adjusted by IMD (model 2a) or SMI diagnosis (model 2b).
*P<0.05, **p<0.01, ***p<0.001.
COPD, chronic obstructive pulmonary disease; CVA, cerebrovascular accident; HoNOS, Health of Nations Outcome Scale; IMD, Index of Multiple Deprivation; SMI, severe mental illness.

and disability within SSD and BD groups. When we examined the independent association of each physical health condition and disability within groups, our results suggested that sociodemographic factors could have a greater impact in these associations in individuals with SSD. Although our results are not directly comparable with previous studies, they are in line with findings in previous research in ageing[62] or some specific SMI populations.[63 64] Further research is needed to understand the potential shared drivers of disability in individuals with BD and these conditions, in general, and diabetes and BD, in particular.

One of the main strengths of this study is the large comprehensive cohort of people with SMI drawn from a population with a high ethnic diversity, addressing the neglect of both ethnic minority groups and SMI in multimorbidity research. This is a key advantage of using EHRs from a large secondary mental healthcare provider and having the benefits of a data extraction strategy to access data on physical health conditions from the text fields of clinical notes. MedCAT development and deployment in CRIS text records will hopefully promote and facilitate future research in mental healthcare. However, further research is needed to validate this strategy in other EHRs sources using free text. Although our results are promising and the comparability of the findings with previous research provides some evidence of validity, further research is also needed to examine the cross-validity using primary care structured fields data. This study is limited to individuals with SMI which does not allow us to compare comorbidity figures of SMI with other common mental disorders and/or the general population. Future studies should replicate our data extraction strategy in other sex and age matched cohorts and explore the potential risks for subsequent health conditions maximising the longitudinal nature of this data source. Furthermore, it is also important to note that individuals with more severe SMI may have more comprehensive textual data. Thus, our findings might be less representative of highly functioning individuals with less severe SMI. In addition, we acknowledge that although the conditions considered are within the most considered in multimorbidity research, future studies should consider a larger number of conditions and include rare diseases and all the conditions needed to calculate standard comorbidity scores such as Charlson or Exlihauser indexes which were not considered in this study (eg, hemiplegia or paraplegia, peptic ulcer disease or AIDS/HIV). It should be noted that our study is one of the first, to our knowledge, to compare the associations between physical health comorbidities and disability in this traditionally neglected population and HoNOS is a widely used measure in secondary mental health services in the UK which provides us a general overview of disability in this population. However, we acknowledge that further research with more objective measures of disability is also needed to drive future policy in this population. To sum up, our study provides an overview of the most prevalent health conditions in SMI and underlines the need for further research into the origins of multimorbidity in this population, considering in more detail the nature of the SMI both in terms of severity and in terms of constituent diagnoses and/or symptomatic phenotype, given the apparent differences between BD and SSD. Our findings highlight multimorbidity as a driver of disability in this population, which also requires further mechanistic evaluation.

**Author affiliations**
[1]Department of Biostatistics and Health Informatics, Institute of Pyschiatry, Psychology and Neurosciences, King's College London, London, UK
[2]NIHR Biomedical Research Centre and Maudsley NHS Foundation Trust, King's College London, London, UK
[3]Department of Psychological Medicine, Institute of Psychiatry, Psychology and Neuroscience, King's College London, London, UK
[4]Health Data Research UK London, University College London, London, UK
[5]Institute of Health Informatics, University College London, London, UK

**Acknowledgements** We would like to thank Megan Pritchard and Mathew Broadbent for their unvaluable contribution and support.

**Contributors** RB conceived and designed the study. ZK, RB, AR and RS designed, validated the data extraction strategy. ZK, TS and AM developed the natural language processing algorithm and interface for the annotations. RB, ZK, JC, LM, NC, TS, AM, NS and TW were annotators. RB, ZK, SS, LL, JC, SA, RD and DB performed the data analyses and/or interpreted the results. RS and JD-M provided clinically relevant input over all the stages. RB, ZK and SS drafted the first version of the manuscript and all authors critically reviewed the manuscript and contributed to writing the final version. RB and RD are responsible for the overall content and guarantors.

**Funding** RB is funded in part by grant MR/R016372/1 for the King's College London MRC Skills Development Fellowship programme funded by the UK Medical Research Council (MRC) and by grant IS-BRC-1215-20018 for the National Institute for Health Research (NIHR) Biomedical Research Centre at South London and Maudsley NHS Foundation Trust and King's College London. JD-M is funded by the Health Foundation working together with the Academy of Medical Sciences, for a Clinician Scientist Fellowship and by the ESRC in relation to the SEP-MD study (ES/S002715/1) and part supported by the ESRC Centre for Society and Mental Health at King's College London (ESRC Reference: ES/S012567/1). DB is funded by a UKRI Innovation Fellowship as part of Health Data Research UK MR/S00310X/. AM is funded by Takeda California. RD, RS and AR are part-funded by the National Institute for Health Research (NIHR) Biomedical Research Centre at South London and Maudsley NHS Foundation Trust and King's College London. RD's work is supported by (1) National Institute for Health Research (NIHR) Biomedical Research Centre at South London and Maudsley NHS Foundation Trust and King's College London; (2) Health Data Research UK, which is funded by the UK Medical Research Council, Engineering and Physical Sciences Research Council, Economic and Social Research Council, Department of Health and Social Care (England), Chief Scientist Office of the Scottish Government Health and Social Care Directorates, Health and Social Care Research and Development Division (Welsh Government), Public Health Agency (Northern Ireland), British Heart Foundation and Wellcome Trust; (3) The National Institute for Health Research University College London Hospitals Biomedical Research Centre.

**Disclaimer** The views expressed are those of the author(s) and not necessarily those of the NHS, the NIHR, the Department of Health, the MRC, ESRC or King's College London. This paper represents independent research funded by the National Institute for Health Research (NIHR) Biomedical Research Centre at South London and Maudsley NHS Foundation Trust and King's College London and by the Health Foundation.

**Competing interests** None declared.

**Patient consent for publication** Not applicable.

**Ethics approval** This project was approved by the CRIS Oversight Committee which is responsible for ensuring all research applications comply with ethical and legal guidelines. The CRIS system enables access to anonymised electronic patient records for secondary analysis from SLaM and has full ethical approvals.

**Provenance and peer review** Not commissioned; externally peer reviewed.

**Data availability statement** Data may be obtained from a third party and are not publicly available. Due to the confidential nature of free-text data, we are unable to make patient-level data available. CRIS was developed with extensive involvement from service users and adheres to strict governance frameworks managed by service users. It has passed a robust ethics approval pro-cess acutely attentive to the use of patient data. Specifically, this system was approved as a dataset for secondary data analysis on this basis by Oxfordshire Research Ethics Committee C (08/H06060/71). The data are deidentified and used in a data-secure format and all patients have the choice to opt-out of their anonymised data being used. Approval for data access can only be provided from the CRIS Oversight Committee at SLaM.

**ORCID iDs**
Rebecca Bendayan http://orcid.org/0000-0003-1461-556X
Leona Leipold http://orcid.org/0000-0002-7712-4293

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
