## [Reviewer comments · BMJ Open]

ARTICLE DETAILS

TITLE (PROVISIONAL)	Mapping Multimorbidity in Individuals with Schizophrenia and Bipolar Disorders: Evidence from the South London and Maudsley NHS Foundation Trust Biomedical Research Centre (SLAM BRC) Case Register
AUTHORS	Bendayan, Rebecca; Kraljevic, Zeljko; Shaari, Shaweena; Das-Munshi, Jayati; Leipold, Leona; Chaturvedi, Jaya; Mirza, Luwaiza; Aldelemi, Sarah; Searle, Thomas; Chance, Natalia; Mascio, Aurelie; Skiada, Naoko; Wang, Tao; Roberts, Angus; Stewart, Robert; Bean, Daniel; Dobson, Richard

VERSION 1 – REVIEW

REVIEWER	Castillejos, M. University of Málaga, Department of Personality, Assessment and Psychological Treatment
REVIEW RETURNED	20-Jun-2021

GENERAL COMMENTS	First of all, I would like to congratulate the authors for the great work they have done. The comorbidity of physical problems in patients with severe mental illness is an issue of special importance, nevertheless on many occasions it has been underestimated and poorly studied. This paper is presented in a well structured way, and each section is adequately explained. Both the tables and the supplementary material are very detailed. The main strength is that it included a large sample and a great number of variables were analysed. The only thing I miss is a discussion about the limitations of the study, since in the section Strengths and Limitation it has not been done.
--

REVIEWER	Baca-Garcia, Enrique University Hospital Jimenez Diaz Foundation, Psychiatry
REVIEW RETURNED	03-Jul-2021

GENERAL COMMENTS	It is a very interesting article with a solid methodology. From the clinical point of view it would be necessary to compare the comorbidity figures of SMI with other common mental disorders and with the general population adjusted for sex and age. I understand that this is likely to be a future work and the interest of the authors is to demonstrate that the method is excellent to correctly establish comorbidity from medical records. I suggest the same analytic approach than Maomen et al (N Engl J Med 2020;382:1721-31). I consider that it is necessary to use the Charlson or Exlihauser index to determine the impact of comorbidity on survival. The calculation of these indexes are easy and I believe that they can give clinicians a better idea of the impact of comorbidity.
--

VERSION 1 – AUTHOR RESPONSE

Reviewer: 1

Dr. M. Castillejos, University of Málaga

Comments to the Author:

First of all, I would like to congratulate the authors for the great work they have done. The comorbidity of physical problems in patients with severe mental illness is an issue of special importance, nevertheless on many occasions it has been underestimated and poorly studied.

This paper is presented in a well structured way, and each section is adequately explained. Both the tables and the supplementary material are very detailed. The main strength is that it included a large sample and a great number of variables were analysed. The only thing I miss is a discussion about the limitations of the study, since in the section Strengths and Limitation it has not been done.

Response: Many thanks for your positive comments. We have now included a more thorough discussion of the limitations of the study to address some of the points raised. We have also added limitations in the strengths and limitations bullet points.

Reviewer: 2

Prof. Enrique Baca-Garcia, University Hospital Jimenez Diaz Foundation, Autonomous University of Madrid

Comments to the Author:

It is a very interesting article with a solid methodology.

From the clinical point of view it would be necessary to compare the comorbidity figures of SMI with other common mental disorders and with the general population adjusted for sex and age. I understand that this is likely to be a future work and the interest of the authors is to demonstrate that the method is excellent to correctly establish comorbidity from medical records. I suggest the same analytic approach than Maomen et al (N Engl J Med 2020;382:1721-31).

Response: We want to thank the reviewer for the positive comments. We agree that it would be interesting to compare the comorbidity figures of SMI with other common mental disorders and with the general population adjusted for sex and age. At this point we have extracted these conditions in the SMI cohort at SLaM and we are starting to validate the data extraction strategy in other cohorts, so we have the plan to look at this in future work. We have now included this in our discussion section as future study, and we are happy to find that we are in the right direction. Many thanks for the very interesting reference. We will study how to incorporate it in our next steps. I consider that it is necessary to use the Charlson or Exlihauser index to determine the impact of comorbidity on survival. The calculation of these indexes are easy and I believe that they can give clinicians a better idea of the impact of comorbidity.

Response: Many thanks for this comment. It is extremely useful, and we will aim to include those conditions that we are missing to compute these indexes in the next data extraction. Unfortunately, annotating, training and validating new conditions takes a very long time and extra-resources which we are not able to have for this piece of work at the moment, but we will plan to include these in the next steps to ensure that we can compute these indexes. We have included this consideration in our limitations.